# Recommendations for the Diagnosis and Therapeutic Management of Hyperammonaemia in Paediatric and Adult Patients

**DOI:** 10.3390/nu14132755

**Published:** 2022-07-02

**Authors:** Amaya Bélanger-Quintana, Francisco Arrieta Blanco, Delia Barrio-Carreras, Ana Bergua Martínez, Elvira Cañedo Villarroya, María Teresa García-Silva, Rosa Lama More, Elena Martín-Hernández, Ana Moráis López, Montserrat Morales-Conejo, Consuelo Pedrón-Giner, Pilar Quijada-Fraile, Sinziana Stanescu, Mercedes Martínez-Pardo Casanova

**Affiliations:** 1Department of Paediatric Inborn Errors of Metabolism, Hospital Universitario Ramón y Cajal, Carretera de Colmenar Viejo, Km. 9, 100, 28034 Madrid, Spain; sinziana.stanescu@salud.madrid.org (S.S.); mmpc00@gmail.com (M.M.-P.C.); 2Department of Adult Inborn Errors of Metabolism, Hospital Universitario Ramón y Cajal, Carretera de Colmenar Viejo, Km. 9, 100, 28034 Madrid, Spain; arri68@hotmail.com; 3Department of Inborn Errors of Metabolism, Hospital Universitario 12 de Octubre, Avenida de Córdoba, s/n, 28041 Madrid, Spain; delia.barrio@salud.madrid.org (D.B.-C.); mgarciasilva@salud.madrid.org (M.T.G.-S.); emartinhernandez@salud.madrid.org (E.M.-H.); pilar.quijadaf@salud.madrid.com (P.Q.-F.); 4Department of Paediatric Nutrition and Metabolic Diseases, Hospital Universitario La Paz, Paseo de la Castellana, 261, 28046 Madrid, Spain; anabergua@gmail.com (A.B.M.); morais.lopez@salud.madrid.org (A.M.L.); 5Department of Gastroenterology and Nutrition, Hospital Universitario Niño Jesús, Avenida de Menéndez Pelayo, 65, 28009 Madrid, Spain; elviracaedo@yahoo.com (E.C.V.); cpedronginar@gmail.com (C.P.-G.); 6Department of Paediatric Nutrition, Centro Médico D-Medical, Calle del Príncipe de Vergara, 44, 28001 Madrid, Spain; maquedalama@gmail.com; 7Department of Internal Medicine, Hospital Universitario 12 de Octubre, Avenida de Córdoba, s/n, 28041 Madrid, Spain; montserrat.morales@salud.madrid.com

**Keywords:** ammonia, hyperammonaemia, diagnosis, therapeutics, urea cycle disorders, haemodialysis

## Abstract

Hyperammonaemia is a metabolic derangement that may cause severe neurological damage and even death due to cerebral oedema, further complicating the prognosis of its triggering disease. In small children it is a rare condition usually associated to inborn errors of the metabolism. As age rises, and especially in adults, it may be precipitated by heterogeneous causes such as liver disease, drugs, urinary infections, shock, or dehydration. In older patients, it is often overlooked, or its danger minimized. This protocol was drafted to provide an outline of the clinical measures required to normalise ammonia levels in patients of all ages, aiming to assist clinicians with no previous experience in its treatment. It is an updated protocol developed by a panel of experts after a review of recent publications. We point out the importance of frequent monitoring to assess the response to treatment, the nutritional measures that ensure not only protein restriction but adequate caloric intake and the need to avoid delays in the use of specific pharmacological therapies and, especially, extrarenal clearance measures. In this regard, we propose initiating haemodialysis when ammonia levels are >200–350 µmol/L in children up to 18 months of age and >150–200 µmol/L after that age.

## 1. Introduction

Ammonia results of the nitrogen metabolism, which is essential for growth and life-maintenance processes. Excess ammonia must be eliminated through the urea cycle due to its high toxicity [1,2]. Hyperammonaemia is defined as levels >110 µmol/L (198 µg/dL) in the neonatal period (including preterm) and >50 µmol/L (90 µg/dL) from that age onwards [1,2,3,4,5]. Ammonia diffuses through all body membranes, including the blood–brain barrier, and alters the amino acid pathway, ion transporters, and neuronal oxide-reduction metabolism, with deleterious effect on neuron and astrocyte functioning [6,7]. Indeed, hyperammonaemia is an emergency condition that may cause severe neurological sequelae and death.

In children, hyperammonaemia is often considered a metabolic derangement due to rare inborn errors of metabolism [1,2,3,4,5]. In adults, it is associated to severe hepatic failure [8,9] and often regarded as an expected finding without damaging capacity. It is not usually acknowledged that patients of any age may exhibit hyperammonaemia secondary to intake of certain drugs, infections, surgeries, or many other conditions (Table 1) [10,11,12], including liver, lung, kidney, or bone-marrow transplantation [13,14,15]. Furthermore, the number of patients with inborn errors of metabolism reaching adulthood has risen due to improvements in diagnosis and treatment in childhood, and the recognition of less severe forms with later onset. Therefore, hyperammonaemia is an all-age condition resulting from heterogeneous causes that many clinicians will eventually have to face over time.

Although several guidelines have approached the diagnosis and management of hyperammonaemia [3,4,5], it is worrisome that patients’ outcomes remain poor [16,17]. We have also noted that most protocols focus on deficiencies in the urea cycle or other metabolic pathways that usually arise in childhood, and do not adequately address hyperammonaemia at older ages or due to non-genetic causes. In 2009, we published the Spanish–Portuguese protocol that was the result of sharing and updating the therapeutic approaches of our Madrid working group and that of Santiago de Compostela, and then agreed with more than fifty specialists from both countries [18]. To our knowledge, the first protocol including specific measures for adults was an update to these recommendations proposed by our working group in 2017 [19].

Here we provide an update to our previously published guidelines that intends to increase the awareness of all physicians of the possibility of hyperammonaemia at any age, as well as providing tools for its diagnosis and early and intensive treatment to achieve a better outcome for our patients. This protocol, summarized in Figure 1, is based on recent publications and the consensus of our group of experts [4,11,13,16,18,19,20].

## 2. Diagnosis of Hyperammonaemia

### 2.1. Clinical Presentation

The severity of the clinical presentation depends on the peak ammonia level but is also influenced by the age of the patient, speed of onset or the presence of other underlying conditions.

The predominant clinical signs of acute hyperammonaemia are mostly due to cerebral oedema, sometimes complicated by multiorgan failure. They include irritability, rejection of feeds, vomiting or drowsiness in neonates and infants. In children, adolescents and adults, the main clinical signs are associated with differing degrees of acute encephalopathy: altered consciousness, ataxia, seizures, and coma [1,2,3,4,5]. In many cases there is a triggering factor such as an infection, surgery, postpartum period, or use of specific medications. 

Persistent or intermittent hyperammonaemia can produce chronic symptoms such as psychomotor delay or growth retardation. Older individuals can exhibit eating disorders or a wide spectrum of neuropsychiatric symptoms. These patients may suffer of recurrent episodes of vomiting, ataxia, lethargy, or behavioural disturbances usually associated with triggering factors. Progressive brain lesions lead to brain atrophy and, in children, microcephaly [16,17,20,21,22].

The broad range of clinical manifestations of hyperammonaemia lead us to emp-hasize that it should be suspected in any patient with neurological or psychological symptoms. Although signs and symptoms associated with triggering causes may also explain the neurological or multiple-organ complications, ruling out concurrent hyperammonaemia is critical to stablish prompt treatment and avoid its serious repercussions.

### 2.2. Diagnostic Approach

Ammonia level determination needs careful blood extraction and sample processing techniques because its levels easily rise when these requirements are not fulfilled (factitious hyperammonaemia). The blood sample can be venous or arterial but needs to be drawn while the corresponding muscle group is at rest, without hypoxia (no compression or holding), and preferably through a large-calibre route to avoid haemolysis. The blood should be kept cold and processed within the next hour. The importance of ammonia measurement and the need for its speedy processing is the reason why its measurement should be available as an urgent determination in all hospital settings. When results do not correspond with the patient’s symptoms, determination should be repeated (Figure 1).

Analysis of the cerebrospinal fluid might seem warranted because of neurological symptoms. We highlight the importance of measuring ammonia before performing a lumbar puncture, since levels over 150 μmol/L entail a high risk of brain wedging due to cerebral oedema.

Other analytical determinations are required to reach a definitive diagnosis, determine the degree of decompensation in already diagnosed patients and/or to assess the presence of a triggering factor that might require specific measures (Table 2). All blood diagnostic samples must be obtained before starting any therapies and urinary samples should be collected as soon as possible. Depending on the determination and the setting, they will be processed immediately or sent to specialized laboratories.

The hospital specialist in metabolic diseases and/or the nearest specialized referral centre must be contacted for guidance. In neonatal cases, the newborn screening results must be requested.

## 3. Management of Hyperammonaemia in the Acute Phase 

We consider the acute phase the time required to reach normal ammonia concentrations, which ideally should occur in the first 24–48 h. All measures taken in this time aim to maintain the patient alive and minimize neurological sequelae. 

Figure 1 details the treatment algorithm according to ammonia levels. Although the aetiology of hyperammonaemia is helpful to determine the need for specific measures, reaching a definitive diagnosis can take several days or longer. As the initial steps to control high ammonia levels are similar in all cases and treatment should not be delayed, these measures are the focus of our protocol. When the patient already has a diagnosis, only the specific nutritional and pharmacological measures for that disease should be used, and the patient’s specialist should be contacted immediately.

The ammonia levels should be measured every 2–4 h in children and at least every 4–8 h in adults throughout this period, together with blood gases and ion levels. If ammonia levels decline, the same treatment measures can be continued until levels normalize. If levels rise, remain without a significant reduction or normalization is not achieved in 12–24 h, adding next-step drugs, or starting extrarenal clearance procedures should be considered.

### 3.1. General and Brain Protection Measures

Patients with an acceptable neurological status can be managed in a regular hospital setting. Oral feeding or administration of drugs may be maintained. Those with decreased consciousness, suspected intracranial hypertension and/or very high ammonia levels require attention in an intensive care unit. Two venous lines, one of them central, and a urinary catheter, are recommended to obtain the diagnostic samples, infuse treatment, and monitor changes. A nasogastric or transpyloric tube can be considered to ensure an adequate nutrient intake and drug administration. Protection measures to treat brain oedema include an elevated head position, sedoanalgesia, and endotracheal intubation with assisted ventilation. Additional treatments for brain oedema should be individualized.

In all cases, triggers for hyperammonaemia or neuronal risk factors should be treated. Drugs that can interfere with the production or elimination of ammonia should be avoided (Table 1). 

### 3.2. Nutritional Treatment

Nutritional treatment is the most important measure to be taken and must begin immediately. To avoid ammonia production through nitrogen metabolism, protein intake must be reduced or suppressed. It is just as relevant to ensure an adequate caloric intake to slow down internal catabolism. Nutritional treatment of acute hyperammonaemia is summarized in Table 3.

If the patient is alert, has good tolerance and moderately high ammonia levels, oral nutrition can be maintained. In these cases, the patient should be encouraged to eat frequent meals, mainly foods with very low protein content. Protein-free supplements are usually necessary to reach caloric requirements and must be adapted to age and clinical situation. Patients with liver failure and low increase in ammonia levels can improve with a low-protein diet that includes branched-chain amino acids, and individuals with metabolic diseases, with the use of specific formulas or supplements. Enteral feedings with nasogastric or transpyloric tubes might be considered in cases in which a continuous feeding regime might prove useful (e.g., to avoid fasting at night) or if the patient does not reach the necessary ingestion orally.

Severe cases need to be managed by parenteral measures. All protein ingestion must be stopped and a perfusion of glucose solutions with ions has to be initiated. We recommend avoiding solutions with a high lactate content until the final diagnosis is made, as it might worsen the situation of certain metabolic pathways. The initial perfusion recommended is a 10% glucose concentration with an infusion rate calculated to cover normal caloric requirements depending on age (Table 3). When a central line is available, a higher glucose concentration can be supplied with less volume, which is especially important if there are signs of cerebral oedema. The perfusion rate should be modified according to diuresis, fluid balance and analytics. It is important to avoid dehydration because it favours hyperammonaemia, but presence of intracranial hypertension must also be considered. Glucose levels must be monitored and if they persist beyond 140–180 mg/dL, the glucose infusion rate should not be decreased but rather an insulin perfusion must be used to normalize glycaemia while promoting anabolism.

It is important to keep in mind the ionic formulation of the drugs used, as benzoate and phenylbutyrate contain sodium. Adjustments will also be needed in patients on diuretics or extrarenal clearance measures (20–25% loss of plasma nutrients during the dialysis).

Complete protein suppression should not be maintained longer than 48 h. If the patient has not normalized ammonia levels by that time, triggering factors, caloric intake, doses of medication and/or intensity of extrarenal depuration should be reassessed. Once ammonia reaches normal levels, the introduction of protein and subsequent dietary treatment should be established. If a diagnosis is available, specific measures can be implemented. Long-term diagnostic (such as genetic) or therapeutic procedures will have to be undertaken by the specialist with experience on the disease suggested by the initial diagnostic results. 

### 3.3. Pharmacological Treatment

Treating exclusively with nutritional measures can be considered in patients with ammonia levels between 50 and 100 μmol/L (up to 150 μmol/L in neonates). With higher levels, if the patient is clearly symptomatic, or if the dietary measures are insufficient, pharmacological treatment should be initiated.

The treatment choice should be based on the symptoms, ammonia levels and, on the most likely diagnosis depending on previous knowledge or urgent laboratory results. Table 4 summarizes the drugs that are useful in the treatment of hyperammonaemia.

Urea cycle stimulation (e.g., L-arginine and N-carbamylglutamate) is useful when the urea cycle is compromised due to secondary reasons, which are the majority of cases. All aetiologies, including primary urea cycle disorders, respond to ammonia clearance through alternative routes (e.g., sodium benzoate, sodium phenylacetate and sodium phenylbutyrate). 

Concurrent infections must be addressed in all patients as a triggering factor and treated with empirical antibiotics. Specific antimicrobial agents and osmotic laxatives have proven to be effective in cases of hepatic insufficiency [23] and propionic acidaemia [24] by reducing overgrowth of ammonia-producing gut bacteria, and they might be useful in other aetiologies. However, these measures can take several days to be effective, and when ammonia levels are elevated, other nutritional and pharmacological treatments should be implemented to accelerate their reduction. Arginine and/or N-carbamylglutamate are useful in most patients, including those with hepatic insufficiency [13].

In patients with an unknown aetiology, cofactors that improve metabolic deficiencies, which secondarily inhibit the urea cycle (e.g., carnitine, hydroxocobalamin and biotin), must be used. In patients with a known diagnosis, only the drugs proven effective for each aetiology should be used, and other specific therapies (citrulline, thiamine, etc.) should be considered. We recommend following the international guidelines published for each disease [3,14,19,24].

The liquids and glucose given with the different medications must be considered when calculating the fluid therapy, which also depends on the time of administration and the patient’s body weight. In cases in which extrarenal measures are added, doses must be adjusted having in mind a 25–50% clearance rate.

### 3.4. Extrarenal Clearance Procedures

The potential need for extrarenal clearance must be considered in all patients with hyperammonaemia. Peritoneal dialysis is hardly effective for ammonia clearance and should not be used. Preparation for haemodialysis is time consuming and might require transfer to another centre. Therefore, in cases of moderate to severe hyperammonaemia, actions concerning this procedure should not be delayed until response to nutritional and pharmacological measures has been assessed.

Our opinion is that haemodialysis should be initiated with levels >200–350 μmol/L in children up to 18 months of age and >150–200 μmol/L after that age. Other indications of extrarenal clearance are an impossibility to begin pharmacological treatment, a lack of a significant response to other therapies, or the presence (or suspicion) of an underlying condition that might also improve with dialysis.

For any patient >1 year of life (>10 kg) haemodialysis with ultrafiltration is the technique of choice because of its greatest efficiency. In younger patients continuous veno-venous or arterio-venous hemodiafiltration or extracorporeal membrane oxygenation with hemodiafiltration are preferred. When these techniques are not available, hemofiltration would be the best option but poses technical problems in very small children. Exchange transfusion is barely effective and only indicated as a temporary measure. The expertise of the clinical team is essential when deciding which of these procedures to apply.

As previously noted, extrarenal clearance causes a significant loss of some essential nutrients and medications and their supply must be increased by 25–50%. 

### 3.5. Poor Neurological Prognostic Factors

Hyperammonaemia affects neurons both by direct interference with their metabolism and by physical compression due to brain swelling. Patients who survive these episodes almost always suffer some kind of long-term neurological sequelae. The prognosis of hyperammonaemia largely depends on age, peak ammonia level, speed of onset and duration of coma. Poor neurological prognostic factors include:Ammonia levels >1000 μmol/L at diagnosis, especially when it increases or does not begin to decrease within the next 12 h.Ammonia levels >2000 μmol/L at any time in the evolution.Ammonia levels >700 μmol/L maintained for more than 48–72 h.Intracranial pressure >30 mm Hg maintained for more than 24 h.Presence of decortication movements.Electroencephalographic signs of brain death.

The presence of one or more of these factors might warrant a limitation of the therapeutic effort due to the high probability of serious and potentially irreversible neurological damage. Other underlying conditions (liver failure, tumours, previous neurological symptoms, etc.) should also be considered. This decision should be made following the centre’s policy regarding these ethical situations and each country´s legal considerations.

## 4. Discussion

In our experience, most physicians regard hyperammonaemia as an uncommon complication due to very infrequent inborn errors of metabolism or late stages of hepatic insufficiency, both of which have a poor intrinsic prognosis that is only slightly further impaired by ammonia elevation. Those of us who have worked in metabolic centres know of the devastating effects of hyperammonaemia at any age and regardless of its cause. The recognition of less severe genetic presentations and mostly the frequent use of certain drugs or surgical procedures that can rise ammonia levels has made apparent that there is a growing group of patients, especially adults, which can suffer greatly from this complication and could have better outcomes if treated correctly (Table 1).

Most published guidelines on hyperammonaemia are directed to metabolic disorders that usually occur in childhood, include very specific diagnostic and therapeutic measures, and do not reach most physicians, especially those treating adult patients [4,6]. We have used these previous guidelines and our own experience to produce a protocol that aims to assist those clinicians who are unfamiliar with metabolic diseases and find the diagnosis and management of hyperammonaemia challenging.

We cannot stress enough the importance of the determination of ammonia in any patient with acute or chronic encephalopathy and/or psychiatric symptoms, regardless of the presence of fever or other conditions that might explain them. If elevated, nutritional, and pharmacological measures should be implemented immediately after collecting the samples for diagnosis. Not only protein suppression but also caloric intake are important (Table 3). Several medications can be used to reduce ammonia levels and should be considered (Table 4), regardless of the underlying aetiology. Time is of essence and frequent monitoring of ammonia levels are indispensable to observe if the measures taken are effective.

Haemodialysis is the best method to rapidly reduce ammonia and other possible pathologic metabolites and its use should not be delayed. Different protocols propose a wide range of ammonia levels as the cut-off points for using this treatment, with much lower levels proposed in adults as proposed in children and adolescents [1,2,3,4,5,23]. Nowadays there is enough experience in the use of these techniques to make them safer and more readily available. Given the deleterious effect of ammonia, it is becoming evident that in order for our patients to have better neurological outcomes, we must promptly begin extrarenal clearance measures [3,25], with levels as low as >200–350 mol/L in children up to 18 months of age and >150–200 mol/L at later ages. An age-dependent difference in tolerance to ammonia, yet to be demonstrated, may be due more to the capacity of the infant´s cranium to expand in the presence of brain oedema thanks to its open fontanel, rather than to differences in brain function or development. That is why we propose the age of 18 months for a change in the indication for dialysis.

As the evolution can be rapid and unpredictable, treatment should be carried out in centres that can provide all possible therapies and have expert clinicians who can manage the diagnostic and therapeutic challenge these patients represent, not only in the acute phase but also in the long term.

## 5. Conclusions

Hyperammonaemia may occur in patients of any age and should be suspected in the presence of acute or chronic encephalopathy and/or psychiatric symptoms. Given its life-threatening effects, we strongly recommend prompt and aggressive intervention, using both a nutritional and pharmacological approach, without delaying extrarenal measures if considered necessary.

## Figures and Tables

**Figure 1 nutrients-14-02755-f001:**
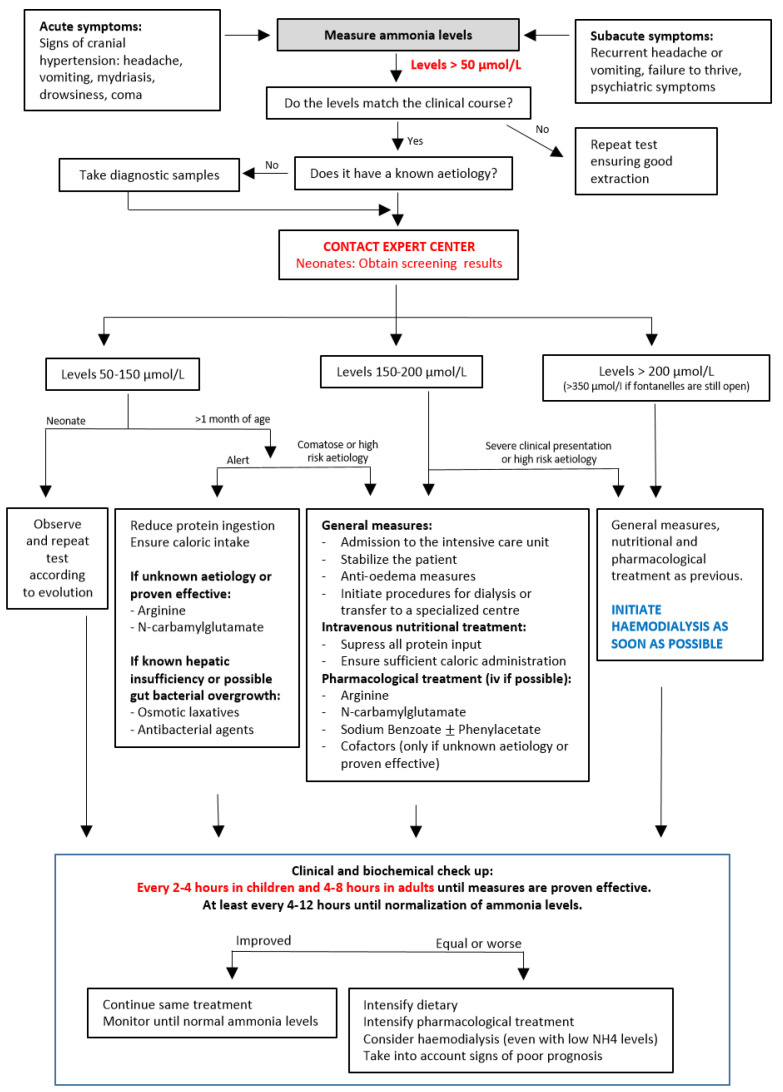
Treatment algorithm for hyperammonaemia (if unknown diagnosis or useful for eathiology).

**Table 1 nutrients-14-02755-t001:** Causes of hyperammonaemia.

Genetic Aetiologies	Non-Genetic Aetiologies	
**Defects of the urea cycle** - N-acetylglutamate synthase (NAGS) - Carbamoyl phosphate synthetase 1 (CPS1) - Ornithine transcarbamylase (OTC) - Argininosuccinate synthase (ASS; citrullinemia type 1) - Argininosuccinate lyase (ASL; argininosuccinic aciduria) - Arginase (ASA; argininemia) - Hyperornithinemia-hyperammonaemia-homocitrullinuria (HHH syndrome) - Citrin (citrullinemia type 2) **Organic Acidemias:** - Propionic acidaemia - Methylmalonic acidaemia - Dibasic aminoacidurias Type 1 Type 2 (Lysinuric protein intolerance) - Isovaleric and 3-methylcrotonic aciduria - Multiple acyl-CoA dehydrogenase (MADD; Glutaric aciduria type II) - 3-hydroxy-3-methylglutaric aciduria - Multiple carboxylase deficiency **Other enzymatic deficiencies:** - Pyrroline-5-carboxylate synthetase - Carbonic anhydrase - Pyruvate carboxylase - Mitochondrial fatty acid β-oxidation - Persistent hyperinsulinemia-hyperammonaemia - Mitochondrial defects - Gyrate atrophy	**Drugs** - Anticonvulsants (valproate, carbamazepine, topiramate, lamotrigine, primidone, zonisamide). - Cancer treatments (5-fluorouracil, cytarabine, vincristine, etoposide, L-asparaginase, cyclophosphamide, sunitinib, rituximab, regorafenib) - Steroids (high doses) - Narcotics - Anaesthetics (enflurane, halothane) - Barbiturates - Haloperidol - Salicylates - Diuretics (acetazolamide) - Ribavirin - Tranexamic acid - Glycine gel for prostate surgery **Surgeries:** - Transplants (liver, lung, kidney, bone marrow) - Gastric surgeries (gastric bypass, bariatric surgery) - Urinary tract surgeries (ureterosigmoidostomy, prostate resection)	**Liver function related** - Severe liver failure - Transient hyperammonaemia of the newborn - Portosystemic shunt - Reye’s syndrome **Nutrition related** - Enteral/ parenteral nutrition (if low in arginine) - Refeeding syndrome - Severe catabolic state severe malnutrition, competitive muscle exercise, prolonged or repetitive seizures multiple myeloma and other tumours **Infections** - Urealytic germ urinary infection - Bacterial overgrowth - Mucositis - COVID **Other** - Shock and/or dehydration - Gastrointestinal bleeding - Distal renal tubular acidosis - Urinary tract dilatation - Alcohol - Hypoglycin intoxication

**Table 2 nutrients-14-02755-t002:** Analytical determinations necessary for diagnosis in patients with hyperammonaemia.

Samples/Determinations	Considerations
**Urgently processed samples**	
Ammonia	Careful extraction: no compression and through a large-calibre route. Keep the tube cold. Process within 1 h.
Ketone bodies	Using a blood reflective device and/or urine test strip.
Blood gases and anion gap	0.3 mL arterial or venous blood
Lactate	Careful extraction: no compression and through a large-calibre route.
Urgent biochemistry profile	Glycaemia, uric acid, urea, creatinine, total proteins, AST, ALT, gamma-glutamyltransferase, creatine kinase, sodium, potassium, chloride, calcium.
Other	Hemogram, coagulation profile, C-reactive protein and procalcitonin
**Samples obtained in the acute phase and sent to a specialized laboratory as soon as possible**	
Blood aminogram and acylcarnitines	Serum or plasma samples (separate from whole blood). Liquid samples might need refrigeration or freezing for their conservation. Dried blood samples when it is not possible to obtain or adequately process liquid samples. Must differentiate into isolated peaks: glutamate, glutamine, alanine, citrulline, arginosuccinate, cystine, lysine and arginine.
Urine aminogram, orotic acid and organic acids	2–10 mL of the first urine obtained. Liquid samples might need refrigeration or freezing for their conservation. Dried blood samples when it is not possible to obtain liquid samples.
Hormone determination (insulin, C peptide and growth hormone)	If concomitant hypoglycaemia.
Bacterial cultures (blood, urine)	To rule out possible triggering infection.
**Samples that can be obtained later but might be considered in an acute phase in cases with bad prognosis**	
Genetic testing samples	Preferably whole blood samples. Dried blood samples when it is not possible to obtain or adequately process liquid samples.

ALT, alanine transaminase; AST, aspartate transaminase.

**Table 3 nutrients-14-02755-t003:** Nutritional treatment of acute hyperammonaemia.

Mild Hyperammonaemia (<150 μmol/L) Alert Patient Good Oral Enteral Tolerance	Severe Hyperammonaemia (>150 μmol/L) Decreased Conscience Level Decreased Tolerance
Reduce protein supply. Provide small amounts of protein-free food (broth, fruits, juices, etc.) frequently. Protein-free caloric supplements might be required to ensure sufficient intake for age. In cases with oral intolerance or a mildly decreased state of consciousness, a nasogastric or gastrostomy tube can be useful. Special dietary formulas can be maintained. Follow individualized emergency nutritional recommendations.	Stop enteral nutrition. Stop protein supply until normal ammonia levels and no longer than 48 h. Ensure sufficient caloric administration. Using a 10% glucose + ions solution perfusion age-related rate of administration would be: 1–12 months: 8–10 mg/kg/min (5–6 mL/kg/h) 1–3 years: 7–8 mg/kg/min (4–5 mL/kg/h) 4–6 years: 6–7 mg/kg/min (3.5–4 mL/kg/h) 7–12 years: 5–6 mg/kg/min (3–3.5 mL/kg/h) Adolescents: 3–5 mg/kg/min (2.5–3 mL/kg/h) Adults: 3–5 mg/kg/min (2–3 mL/kg/h) Neonates: adequate for age fluid solution with 10–12 mg glucose/kg/min. If possible (available central line), consider a higher glucose concentration and less volume. Consider an insulin perfusion (0.05–0.2 U/kg/h) if persistent glucose levels >140–180 mg/dL.

**Table 4 nutrients-14-02755-t004:** Pharmacologic treatment of acute hyperammonaemia.

Drug	First Dose	Maintenance	Considerations
**Urea cycle enhancers**: useful in cases of hyperammonaemia due to any cause but primary urea cycle deficiencies. Should be always included in cases of unknown aetiology.			
N-carbamylglutamate	100 mg/kg	100–250 mg/kg/day in 2–4 doses	Oral (or crushed through feeding tube): tablets.
			Useful in most genetic and non-genetic disorders. Not useful in most known primary urea cycle disorders (only NAGS deficiency).
			Maximum dose not stablished. In adults use weight for lean body mass.
L-Arginine	<20 kg: 250–400 mg/kg	<20 kg: 250 mg/kg/day	Oral: powder, sachets. IV: diluted in 10% glucose solution. Can be administered together with benzoate.
	>20 kg: 250 mg/kg (max 12 g)	>20 kg: 200 mg/kg/day (max 12 g/day)	

**Urea cycle scavengers**: useful in all cases of hyperammonaemia. Should be included in hyperammonaemia due to primary urea cycle disorders or severe cases of unknown aetiology.			
Sodium benzoate ± Sodium phenylacetate	<20 kg: 250 mg/kg	<20 kg: 250–500 mg/kg/day	IV: requires central venous access.
	>20 kg: 5.5 g/m^2^ (max 12g)	>20 kg: 5.5 g/m^2^ (max 12g/day)	Diluted in 10% glucose solution and administered over 2 h.
			Attention to the sodium content.
			Precaution in organic acidaemias.
Phenylbutyrate		<20 kg: 250–500 mg/kg/day in 4 doses	Oral: tablet, powder, or solution presentations.
		>20 kg: 9.9–13 g/m^2^/day in 4 doses	Slow action: not first option in acute hyperammonaemia.
**Cofactor therapy**: useful if unknown aetiology or an underlying genetic disease is suspected. If known diagnosis, only start those that have been proven effective.			
L-Carnitine	50 mg/kg	100 mg/kg/day in 4 doses	Oral: 10 or 30% solutions.
	Maximum dose: 4 g	Maximum dose: 6 g/day	IV: 20% solution.
			Caution in long-chain fatty acid oxidation deficiencies.
Biotin	10 mg	20–40 mg/day	Oral or iv presentations.
Hydroxocobalamin	1 mg	Repeat only depending on diagnosis	IM or IV.
			Only one dose required initially.
**Therapies aimed to reduce ammonia gut production**: proven effective in cases of hepatic encephalopathy and propionic acidaemia. Slow action: not useful as monotherapy and if high ammonia levels.			
Osmotic laxatives	Lactulose 15–20 mL every 12 h		Titrate until 2–3 stools/day.
	Polyethylene glycol 1 dose		
Antimicrobial agents	Rifaximin 2–12 years of age (off-label) 20–30 mg/kg/day 2–4 doses >12 years of age 200–400 mg/day 2–4 doses		Use preferably antibiotics with low absorption rates. Other options: metronidazole, ciprofloxacin, doxycycline, etc.

## Data Availability

Not applicable.

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
