# Peer review of "Recommendations for the Diagnosis and Therapeutic Management of Hyperammonaemia in Paediatric and Adult Patients"

_nutrients, 2022, doi:10.3390/nu14132755_

Round 1
Reviewer 1 Report
The authors submit a review article. This is an interesting article for the professional public on the topic “recommendations for the diagnosis and therapeutic management of hyperammonemia in pediatric and adult patients”. The article is clearly and comprehensibly written and logically structured. The review is based on recent papers.
Reminders:
Figure 1: poor resolution
Table 2: glucose instead of glycaemia
L, D: capital letters
Author Response
Thank you for your positive review. The minor changes suggested have been undertaken. We will submit a higher resolution copy of Figure 1.
Reviewer 2 Report
This is a clinically important topic. However, the article is not well written. The article sounds like a guide for young doctors developed at some clinic. Many citations are missing.
I'm not sure if the article is suitable for Nutrients. Which nutrients does the article cover? Wouldn't it be more appropriate for a journal focused on hepatology?
Author Response
Dear Reviewer,
This is indeed a guide for doctors, young and old, which might be unaware of the wide range of patients that can be affected by hyperammoniemia or with little experience in its treatment. It is our experience that many physicians regard hyperammoniemia as an infrequent complication (save in hepatic patients). One of the objectives of this paper is to remind physicians of all specialties that this can be a life-threatening complication that is not restricted to hepatic patients, but can also appear in individuals of all ages that use certain medications, undergo surgery, suffer infections or need transplants, among others. Therefore, a wide scope journal such as Nutrients is preferable than a very specialized journal focused on only one of the predisposing conditions.
Nutrients was also selected because this complication can and should be treated, and the most important measure is the dietary intervention. This intervention is explained on the paper. Specific dietary measures dependent on the aetiology are not included because they are not necessary in an acute situation and because the broad range of possibilities greatly exceed the length and purpose of this manuscript.
We want to remind our readers that it is not an unimportant or hopeless complication, but that in order to avoid complications an intervention should be made as soon as possible, by any attending physician. There is no time to wait for an expert. Our focus was not on explaining all the physiopathological pathways involved (usually regarded as difficult and obscure), but on encouraging all physicians to take the first very important steps in treatment. That is why our wording was simple and we avoided in-depth explanations. As you correctly pointed out, "it could be a guide for young doctors developed in some clinic". The sad reality is that in most clinics there is no such guide and patients die or become disabled because of it and this is the gap we want to fill.
Your are indeed correct in that many more citations could be included. Apart from the restrictions set by the journal, we purposely reduced the number of papers referenced that were focused exclusively in children or in certain diseases, as we wanted to have this paper convey the broad range of ages and diseases (no only inborn errors of metabolism or hepatic disease) that can be affected by this complication. We tried to include the most recent and relevant ones or those that supported this idea.
Thank you for your time and your suggestions. It seems we were not able to adequately transmit the objectives of our paper to you. We will modify the manuscript in order to make our ideas clearer and we will go over our references to see if the addition of further citations can be helpful. Hopefully you will find our modifications adequate in your further review.
Reviewer 3 Report
This is a well written and comprehensive rewiev of the treatment and diagnostic work up of patients with hyperammonemia.
It constitutes an excellent help to clinicians who encounter these patients.
A couple of thoughts:
The neonate with galloping hyperammonemia is a difficult challenge and should always result in prompt contact with a specialist centre. Maybe this should be stressed more clearly in Fig 1. The glucose infusion rate for neonates is left out in Table 3 - in line with this.
Under 3.4 Extrarenal clearance procedures: The rate of the increment of the ammonia level is an important guide for the decision to start hemofiltration. You write very clearly that one should start to get ready for hemofiltration / hemodialysis at an early stage and start the treatment at lower ammonia levels than proposed in some earlier publications. This agrees well with our experience that we are often too late - because of too high ammonia levels when they are discovered - but also caused by hesitation. Maybe this could be mentioned to stress this even more.
A couple of details: Table 1: Isovaleric acidemia is mentioned twice in column one. MSUD seldom gives hyperammonemia and if it does it is mild. Table 4: First line concerning N-cabamylglutamate: "Not useful in known urea cycle defects" ought to be "…most known…" Line 228: N-carbamylglutamate instead of N-acetylglutamate
Author Response
Dear Reviewer,
Thank you for your positive comments.
Hyperammoniemia in the neonatal period is usually due to inborn errors of metabolism and there are several specific guidelines regarding these diseases. Also, neonatologists are more aware of this complication. Our main objective was to point out that hyperammoniemia is not as uncommon at other ages and due to other reasons. However, you are totally right in that neonates should not be excluded from our guide for this reason and we will amend the text and figures accordingly.
The early implementation of extrarenal clearance measures is one of our main contributions with this manuscript. As you say, hesitation on when to begin seems like the most important factor in the delay in treatment. We were unsure on how this idea would be considered by other colleagues but your comment helps and we will re-word this section to be more forceful.
The tables and text will be corrected following your suggestions.
Thank you for your time and helpful comments.
Round 2
Reviewer 2 Report
I confirm the previous rating. The article does not sound scientific and lacks many citations. There is no mention of any component of nutrition that Nutrients should address.